# Luteolin Isolated from *Juncus acutus* L., a Potential Remedy for Human Coronavirus 229E

**DOI:** 10.3390/molecules28114263

**Published:** 2023-05-23

**Authors:** Asma Hakem, Lowiese Desmarets, Ramla Sahli, Rawen Ben Malek, Charline Camuzet, Nathan François, Gabriel Lefèvre, Jennifer Samaillie, Sophie Moureu, Sevser Sahpaz, Sandrine Belouzard, Riadh Ksouri, Karin Séron, Céline Rivière

**Affiliations:** 1Joint Research Unit 1158, BioEcoAgro, Univ. Lille, INRAE, Univ. Liège, UPJV, JUNIA, Univ. Artois, Univ. Littoral Côte d’Opale, ICV—Institut Charles Viollette, 59650 Villeneuve–d’Ascq, France; asma.hakem.etu@univ-lille.fr (A.H.); sahliramla@gmail.com (R.S.); rawen.ben-malek@inrae.fr (R.B.M.); gabriel.lefevre@univ-lille.fr (G.L.); jennifer.samaillie@univ-lille.fr (J.S.); sophie.moureu@gmail.com (S.M.); sevser.sahpaz@univ-lille.fr (S.S.); 2Laboratory of Aromatic and Medicinal Plants, Biotechnology Centre of Borj-Cedria (CBBC), Hammam-Lif 2050, Tunisia; ksouririadh@gmail.com; 3Center for Infection and Immunity of Lille (CIIL), Univ. Lille, CNRS, INSERM, CHU Lille, Institut Pasteur de Lille, U1019—UMR 9017, 59000 Lille, France; lowiese.desmarets@ibl.cnrs.fr (L.D.); charline.camuzet@pasteur-lille.fr (C.C.); nathanfrancoislille@gmail.com (N.F.); sandrine.belouzard@ibl.cnrs.fr (S.B.); karin.seron@ibl.cnrs.fr (K.S.)

**Keywords:** HCoV-229E, SARS-CoV-2, MERS-CoV, *Juncus* species, bioguided fractionation, luteolin

## Abstract

The COVID-19 pandemic, caused by SARS-CoV-2, addressed the lack of specific antiviral drugs against coronaviruses. In this study, bioguided fractionation performed on both ethyl acetate and aqueous sub-extracts of *Juncus acutus* stems led to identifying luteolin as a highly active antiviral molecule against human coronavirus HCoV-229E. The apolar sub-extract (CH_2_Cl_2_) containing phenanthrene derivatives did not show antiviral activity against this coronavirus. Infection tests on Huh-7 cells, expressing or not the cellular protease TMPRSS2, using luciferase reporter virus HCoV-229E-Luc showed that luteolin exhibited a dose-dependent inhibition of infection. Respective IC_50_ values of 1.77 µM and 1.95 µM were determined. Under its glycosylated form (luteolin-7-*O*-glucoside), luteolin was inactive against HCoV-229E. Time of addition assay showed that utmost anti-HCoV-229E activity of luteolin was achieved when added at the post-inoculation step, indicating that luteolin acts as an inhibitor of the replication step of HCoV-229E. Unfortunately, no obvious antiviral activity for luteolin was found against SARS-CoV-2 and MERS-CoV in this study. In conclusion, luteolin isolated from *Juncus acutus* is a new inhibitor of alphacoronavirus HCoV-229E.

## 1. Introduction

Viruses are responsible for 25.4% of infectious diseases affecting human health [1]. Some of these viruses are implicated in respiratory infections causing thereby a high rate of mortality worldwide, especially in developing countries. The most well-known examples are the Spanish flu (H1N1) which occurred in Spain in 1918 and caused the death of about 50 million persons worldwide mainly due to bacterial pneumonia complications, as well as the Asian flu (H2N2) in 1957 and the Hong Kong flu (H3N2) in 1968 responsible for the death of 4 million and one million people, respectively. Over the past two decades, three highly pathogenic human coronaviruses belonging to the genus Betacoronavirus have emerged, SARS-CoV (severe acute respiratory syndrome coronavirus) and MERS-CoV (Middle East respiratory syndrome coronavirus), which were first at the origin of two new severe respiratory epidemics [2] and lately SARS-CoV-2.

The SARS-CoV originated in Southern China in 2002 has infected 8437 individuals leading to 813 deaths [3]. The MERS-CoV appeared 10 years later. This virus was isolated, for the first time, from a sputum sample of a male patient suffering from a severe pneumonia in Djeddah (Saudi Arabia) [4]. According to the World Health Organization’s data, since 2012, MERS-CoV has killed 945 people with a case fatality rate of more than 30%, and the epidemic is still ongoing [5]. In December 2019, SARS-CoV-2 was identified as the infectious agent responsible for the COVID-19 pandemic. So far, it is estimated to have caused the death of more than 6.9 million people [6].

Before the emergence of these highly pathogenic coronaviruses, other human coronaviruses existed but not much attention was accorded to them because of their minor pathogenic potential. The human coronavirus 229E (HCoV-229E) was isolated in the mid-1960′s and was originally discovered in Great Britain from a nasal wash of patients with cold [7]. HCoV-229E is an alphacoronavirus characterized by a single positively-charged RNA strand of about 27.7 kb [8]. Its emergence into the human population is estimated to about 200 years ago. HCoV-229E as well as other coronaviruses causing mild symptoms, notably HCoV-OC43, HCoV-NL63, and HCoV-HKU1, are well adapted to humans and do not seem to need an animal reservoir for transmission unlike the betacoronaviruses MERS-CoV, SARS-CoV and SARS-CoV-2. However, the latter now also shows a very efficient human-to-human transmission. Epidemics that have appeared during these last 20 years highlight the ability of coronaviruses to cross barriers between species causing severe pathologies in humans. This clearly underlines that this family of viruses constitutes a reservoir of emerging pathogens [9,10,11].

To deal with the SARS-CoV-2 pandemic, the scientific community worldwide put into action many strategies to counteract SARS-CoV-2 infection, such as blocking SARS-CoV-2 entry with monoclonal antibodies [12], impairing SARS-CoV-2 replication with polymerase or protease inhibitors [13,14], inhibiting excessive inflammatory response by repurposing already existing antivirals and broad-spectrum drugs [15,16]. While RNA vaccines have shown evidence of efficacy to prevent severe outcomes in patients, few other therapeutic options seemed to significantly reduce the symptomatology caused by SARS-CoV-2. Some therapeutic and prophylactic options to help fight COVID-19 were nonetheless developed and approved by the European Medicines Agency (EMA) and/or the Food and Drug Administration via the Emergency Use Authorizations (FDA EUAs), like for instance Monoclonal antibody treatments (mAb) and the new approved drug, Paxlovid^®^.

The efficacy of mAbs treatment against SARS-CoV-2 was demonstrated in several studies including clinical trials, however the anticipated activity of the different anti-SARS-CoV-2 mAb therapies varied dramatically depending on the circulating variant [17,18]. The antiviral drug Paxlovid^®^ is prescribed once the first signs of COVID-19 infection appear to help prevent severe outcomes. This medicine was authorized for marketing across the European Union by the European Commission on the 28th of January 2022. Paxlovid^®^ is composed of 2 drugs: nirmatrelvir, which is an orally bioavailable viral protease Mpro inhibitor and ritonavir, an inhibitor of cytochrome P450 (CYP) 3A4 and a pharmacokinetic boosting agent. Paxlovid^®^ has proven its efficacy clinically in reducing the mortality rate [19]. Two other SARS-CoV-2 antivirals targeting the viral RNA-dependent RNA polymerase are available for COVID-19 therapy, remdesivir and molnupiravir. However, remdesivir benefits are low. Only a small effect against death or progression to ventilation was observed in hospitalized patients that had not yet been ventilated. Recent data on molnupiravir, whose mode of action apparently involves a two-step mechanism, showed no effect on death outcome of the patients [20,21].

Despite the development of highly efficient vaccines, the global fast-spreading of SARS-CoV-2 led to the emergence of multiple new variants decreasing the efficacy of both vaccines and mAb therapy. Therefore, it is clear that vaccines are part of the solution but the search for specific antiviral treatments remains challenging. In this context, our study aims to explore the plant kingdom in order to find specific antiviral natural compounds against coronaviruses, in particular against HCoV-229E, SARS-CoV-2 and MERS-CoV.

Natural products from plant species continue to hold a powerful position in drug discovery process [22]. They are renowned for their structural diversity and their promising biological activities. 42% of the total antiviral agents produced between 1981 and 2019 were derived from natural products [22,23]. So far, a number of high-throughput in silico studies highlighted the great potential of phytochemicals against coronaviruses [24,25]. In the case of our study, we are mainly seeking in vitro evidence. The choice of plant species was based on an ecological approach and we specifically selected halophytes. Even though the definition of halophytes remains debated, these plant species can simply be defined as having the ability to survive in saline environments with NaCl concentrations around 85 mM or more (up to 2 M in the case of some *Tecticornia* species) [26,27,28,29]. In a previous collaborative study, we demonstrated that dehydrojuncusol, a natural phenanthrene derivative, isolated from *Juncus maritimus* was able to inhibit hepatitis C virus replication by targeting the non-structural protein NS5A [30,31]. To conduct our study, different *Juncus* species, halophytes and glycophytes, were selected and collected from different locations in France. Bioguided fractionation was performed in order to thoroughly investigate their antiviral activity against HCoV-229E, SARS-CoV-2 and MERS-CoV coronaviruses and identify the natural bioactive compound(s).

## 2. Results

### 2.1. Antiviral Activity of Different Juncus Species against HCoV-229E

A primary screening based on the use of a luciferase recombinant HCoV-229E (HCoV-229E-Luc) was carried out on the crude methanolic extracts of rhizomes, stems and inflorescences of the collected *Juncus* species (*Juncus acutus* L., *Juncus inflexus* L. and *Juncus maritimus* Lam.) in order to identify the most active extract(s) against HCoV-229E. Infection was performed in Huh-7 cells expressing the transmembrane serine protease 2 (TMPRSS2). TMPRSS2 is a key enzyme in HCoV-229E and SARS-CoV-2 entry as it allows the fusion of the viral particle to the cell membrane, which is the major entry pathway for coronaviruses [32,33,34]. In the absence of TMPRSS2, the virus enters the cell via endocytosis and fusion with the endosomal membrane. Among all crude methanolic extracts tested at 100 µg·mL^−1^ (Figure 1), the highest antiviral activities against HCoV-229E-Luc were respectively observed for *J. acutus* closed inflorescences (locality 1) = JA1 CICE, *J. acutus* rhizomes (locality 2) = JA2 RCE, *J. acutus* open inflorescences (locality 1) = JA1 OICE, *J. acutus* stems (locality 2) = JA2 SCE, and finally for *J. inflexus* rhizomes (locality 3) = JI3 RCE. The activity of these five crude extracts was quite similar.

### 2.2. Antiviral Activity of Sub-Extracts of Juncus acutus Stems (JA2) against HCoV-229E

Ecological reasons to maintain biodiversity led us to continue our study on *Juncus acutus* stems. Indeed, *Juncus* inflorescences were not the most abundant organs in terms of weight of plant raw materials. Even though their crude methanolic extracts demonstrated the most important antiviral activity, it was difficult to continue working on them. Moreover, in an effort to preserve *Juncus* plants, we also decided not to carry on the antiviral study on *J. acutus* rhizomes and *J. inflexus* rhizomes because this plant part is implicated in the vegetative replication of these species and therefore their durability. It was therefore more appropriate to perform our study on the stems of *Juncus acutus* that was collected in the locality 2 in Brittany.

In order to identify the molecule(s) responsible for this activity, bioguided fractionation was performed. At first, liquid-liquid extraction was done using solvents with increasing polarity. Three sub-extracts, methylene chloride (CH_2_Cl_2_), ethyl acetate (EtOAc) and aqueous were obtained and they were, in their turn, evaluated for their antiviral activity against HCoV-229E. A crude methanolic extract of *Mallotus oppositifolius* was used as a control because this latter was previously described as a highly active extract against HCoV-229E [35].

Our results showed that crude methanolic extract, as well as EtOAc and aqueous sub-extracts obtained from JA2, tested at 25 µg·mL^−1^, were the most active against HCoV-229E (Figure 2). EtOAc sub-extract exhibited the most anti-HCoV-229E activity followed by the crude methanolic extract and aqueous sub-extract.

### 2.3. Analysis by UHPLC-UV-MS of Sub-Extracts of Juncus acutus Stems (JA2)

Taking into account the polarity nature of both EtOAc and aqueous sub-extracts, it can be assumed that the compound(s) responsible for this anti-HCoV-229E activity are moderately polar constituents or polar constituents. The existing literature has highlighted the fact that flavonoids are specialized metabolites rarely isolated from *Juncus* species [36]. However, some flavones, including luteolin and some of its derivatives, have already been reported in some *Juncus* species, such as *Juncus acutus* [30,36,37,38].

The different sub-extracts obtained from JA2 were analyzed by UHPLC-UV-MS (Figure 3). The chromatogram of the EtOAc sub-extract at λ = 254 nm showed the presence of two major peaks with the corresponding retention times Rt = 3.42 min and Rt = 4.36 min. Mass spectra for these two peaks gave molecular ions [M − H]^−^ at *m*/*z* 447 and 285, respectively. The comparison of retention times and mass spectra of these two peaks with reference standards confirmed the presence of luteolin (Rt = 4.36 min; MW = 286.24 g·mol^−1^; C_15_H_10_O_6_) and one of its glucoside, luteolin-7-*O*-glucoside (Rt = 3.42 min; MW = 448.4 g·mol^−1^; C_21_H_20_O_11_). These two flavones could potentially be responsible for the antiviral activity highlighted for the EtOAc sub-extract against HCoV-229E. They were also detected in the aqueous sub-extract in trace amounts, which may explain the activity close to that of the crude extract and slightly higher than that of the non-active CH_2_Cl_2_ sub-extract. The chromatogram of this latter, obtained by UHPLC-UV-MS, showed the presence of phenanthrene derivatives as previously detected in other *Juncus* species, such as juncusol. This class of phytochemicals was already reported in other *Juncus* species [39,40].

### 2.4. Identification of the Active Antiviral Compound(s) from the Ethyl Acetate and the Aqueous Sub-Extracts of Juncus acutus Stems (JA2) following Bioguided Fractionation

In the light of the above promising antiviral activity results against HCoV-229E, it was decided to pursue bioguided fractionation process on both EtOAc and aqueous sub-extracts to identify the compound(s) responsible for this activity.

#### 2.4.1. Antiviral Activity of Fractions Obtained from JA2 EtOAc Sub-Extract by Semi-Preparative HPLC

The fractionation of the EtOAc sub-extract of JA2 was performed by semi-preparative HPLC. The 12 pre-purified fractions were subjected to the above-mentioned antiviral activity assay against HCoV-229E. For this experiment, a lower range of concentrations was tested, 10 and 25 µg·mL^−1^.

Our results (Figure 4) clearly demonstrated that fraction F7 showed the most remarkable antiviral activity compared to the rest of the tested fractions, with a reduction of approximatively 2 log_10_ of the relative luciferase activity at 10 µg·mL^−1^.

UHPLC-UV-MS analysis were performed on the 12 fractions. The comparison of their UV chromatograms at λ = 254 nm with those of reference standards (luteolin and luteolin-7-*O*-glucoside) (Figure 5 and Appendix A) highlighted the abundant presence of luteolin in the active fraction F7. On the basis of the PDA chromatogram, its estimated purity was about 92% according to the integration algorithm ‘Apex Track’ of the software Empower^®^ version 3. Fraction F6 appeared to also contain luteolin but in a lesser amount, as well as other minor compounds. The other fractions were slightly active or inactive. Chromatograms from F8 to F12 displayed the presence of some phenanthrene derivatives such as juncusol C_18_H_18_O_2_ (Rt = 5.88 min; MW = 266.3 g·mol^−1^). The fractions F1 to F5 were mainly consisted of luteolin-7-*O*-glucoside.

All these findings supported the plausibility of our hypothesis that luteolin (Figure 6) was the flavone responsible for the antiviral activity against HCoV-229E. The glycosylated form of luteolin, luteolin-7-*O*-glucoside, was not active against HCoV-229E.

#### 2.4.2. Antiviral Activity of Fractions Obtained from JA2 Aqueous Sub-Extract by CPC

Compared to the EtOAc sub-extract, the aqueous sub-extract of JA2 displayed to a lesser degree an antiviral activity against the coronavirus HCoV-229E. Similarly to the EtOAc sub-extract, we wanted to identify the compound(s) responsible for this activity. For this purpose, the aqueous sub-extract was subjected to CPC fractionation using a ternary system composed of EtOAc/Isopropanol/Water (7:3:10). The 12 pre-purified fractions were grouped on the basis of similar chromatographic profiles, then were evaluated for their antiviral activity against HCoV-229E at a concentration of 50 µg·mL^−1^. A higher concentration than the one used for the EtOAc fractions was used because no antiviral activity was observed at 25 µg·mL^−1^.

According to our results, fractions F3 and F4 showed sequentially the highest anti-HCoV-229E activity (Figure 7). As we have previously postulated that luteolin present in both EtOAc and aqueous sub-extracts of *Juncus acutus* stems was responsible for the observed antiviral activity, we analyzed therefore the phytochemical composition of these two active pre-purified fractions by UHPLC-UV-MS. As suggested, our data (Figure 8) clearly confirmed the presence of luteolin in fractions F3 and F4. Hence, the abundance of luteolin in fraction F3 can be correlated with the antiviral activity previously observed for the same fraction. These results are in agreement with those obtained for the EtOAc sub-extract.

Considering all acquired data, luteolin is undoubtedly the compound responsible for the antiviral activity against HCoV-229E of the two JA2 sub-extracts (EtOAc and aqueous). Further experiments were required in order to probe the antiviral activity of luteolin against HCoV-229E and to gain more insight into its toxicity and its potential mechanism of action.

### 2.5. Antiviral Activity of Luteolin against HCoV-229E and Toxicity in Huh-7 Cells

To confirm that luteolin is the active compound identified in the different pre-purified active fractions, its toxicity and its antiviral activity were tested in dose-response experiments. For toxicity, Huh-7 cells were incubated with luteolin at different concentrations for 24 h and the toxicity was quantified by MTS assay (Figure 9A). Our results showed that the half-maximal cytotoxic concentration (CC_50_) of luteolin was approximately 33 µM. For antiviral assays, Huh-7 cells expressing or not TMPRSS2 were inoculated with increasing concentrations of luteolin. Our data showed a dose-dependent inhibition of HCoV-229E-Luc infection confirming that luteolin was the active compound of JA2 sub-extracts (Figure 9B). The IC_50_ was similar in Huh-7 and Huh-7/TMPRSS2 cells (1.77 and 1.95 µM, respectively) underlying thereby that luteolin exhibited an antiviral activity against HCoV-229E regardless the entry pathway. From these data, respective selectivity index (SI) values, 18.6 and 16.9, were determined. The results presented in Figure 9A clearly showed that luteolin was not toxic at active concentration.

### 2.6. Mechanism of Action of Luteolin against HCoV-229E

To determine which virus infection step was inhibited by luteolin, a time of addition assay was performed. The molecule was added, at a concentration of 10 µM, at different time points during infection (Figure 10A). Three different inhibitors were added as control, camostat, a TMPRSS2 inhibitor that inhibits entry, GC376, a protease inhibitor that inhibits replication, and remdesivir, a RdRp inhibitor that also inhibits replication. As shown in Figure 10B, luteolin was the most active when it was added at post-inoculation (PI): either 1 h PI or 2 h PI, similarly to results obtained with GC376 and remdesivir. Taken together, these data indicated that luteolin was an inhibitor of the replication step of HCoV-229E.

### 2.7. Antiviral Activity of Luteolin against SARS-CoV-2 and MERS-CoV

Regarding the strong activity of luteolin on HCoV-229E, the antiviral activity of this molecule was tested against highly pathogenic coronaviruses SARS-CoV-2 and MERS-CoV. GC376, a protease inhibitor or remdesivir, were added as controls. The results showed that GC376 and remdesivir inhibited SARS-CoV-2 and MERS-CoV, respectively. Unfortunately, no inhibition of infection was observed against these two viruses showing that luteolin was not active neither on SARS-CoV-2 nor on MERS-CoV. Altogether, the findings showed that luteolin is a specific replication inhibitor of HCoV-229E but has no obvious antiviral activity on SARS-CoV-2 and MERS-CoV in vitro (Figure 11).

## 3. Discussion

Human coronaviruses, belonging to the genus Alphacoronavirus or Betacoronavirus, have the ability to trigger symptoms with different severity; from common cold to life-threatening respiratory illnesses in the lower respiratory tract [41]. The recent outbreak of COVID-19 has pointed out the lack of clinically proven therapeutics to combat coronavirus infections. Plants are known to be a great source of antimicrobial agents [22,42]. They can be selected according to different approaches: traditional knowledge, chemotaxonomy or ecological criteria. In a previous study conducted in our laboratory, we demonstrated the relevance of the ecological approach by underlining the antiviral activity of *Juncus maritimus* against hepatitis C virus and highlighting the potential of dehydrojuncusol as a NS5A inhibitor [30,31]. In this present study, we continued to investigate the antiviral potential of *Juncus* species, whether halophytes and glycophytes, but this time against coronavirus infections. Considering some aspects of biodiversity preservation and accessibility to resources, we have chosen to deepen the antiviral activity of the crude methanolic extract of *Juncus acutus* stems, even if other crude extracts were slightly more active (inflorescences or rhizomes). HCoV-229E was selected for the evaluation of the antiviral activity of the selected plants because of its minimal pathogenicity and its general association with asymptomatic to mild disease and sometimes acute respiratory distress syndrome [43]. Bioguided fractionation, combining conventional purification techniques (liquid-liquid extraction, semi-preparative HPLC and CPC) with in vitro antiviral experiments allowed us to identify the molecule, present in the crude methanolic extract, responsible for this activity. It is the flavone luteolin or 3′,4′,5,7-tetrahydroxyflavone. This specialized metabolite was the major compound of the active fraction F7 obtained from the ethyl acetate sub-extract of *J. acutus* stems. Luteolin is commonly ubiquitous throughout the plant kingdom and was described as one of the major phenolic flavonoids of soft rush (*Juncus effusus* L.) [44,45]. This flavone is frequently present under its glycosylated form in various dietary sources and medicinal plants. The health benefits of luteolin have been extensively described in literature, among them, antioxidant, anti-inflammatory, sun protectant, antimicrobial, chemoprotective and chemotherapeutic properties [46,47,48,49]. In the case of our antiviral study, luteolin was inactive against HCoV-229E under its glycosylated form, luteolin-7-*O*-glucoside, the main constituent of fractions F2, F3, F4 and F5 of the EtOAc sub-extract and fractions F5 and F6 of the aqueous sub-extract of *J. acutus* stems (JA2). Likewise, phenanthrene derivatives, identified in the methylene chloride sub-extract, which had previously shown an antiviral activity against hepatitis C virus, were not active against coronaviruses.

Before conducting our antiviral study against HCoV-229-E and SARS-CoV-2, it was important to ensure that luteolin does not affect cellular functions when administered at the active dose. In this study, we determined a half-maximum cytotoxic concentration (CC_50_) equal to 33 µM for luteolin using MTS assay on Huh-7 cells. A higher CC_50_ (155 µM) was reported on Vero E6 cells by the MTT assay [50]. In another study, luteolin showed a cytotoxic effect at 50 µM on MH14 cells, which are a derivative of the Huh-7 cell line [51].

Afterward, we determined the antiviral activity of luteolin against HCoV-229E using Huh-7 and Huh-7/TMPRSS2 cells. Similar IC_50_ values were obtained in both cell types, 1.77 and 1.95 µM respectively. These results showed clear evidence that luteolin exhibits anti-HCoV-229E activity regardless the entry pathway.

The antiviral activity of luteolin has been highlighted in many studies against both enveloped and non-enveloped viruses; among them the Influenza A virus, the Japanese encephalitis virus, the hepatitis B virus, the human immunodeficiency virus (HIV-1), the enterovirus 71 (EV71) and the coxsackievirus A16 (CA16) [49,52,53,54,55].

Depending on the studied virus, different mechanisms of action were reported for luteolin. For enteroviruses EV71 and CA16, it was reported that luteolin targets their post-attachment stage infection and block their RNA synthesis [55]. Luteolin was also described as anti-HIV-1 molecule, precisely at the Tat-LTR transactivation (transcriptional step) [54]. Moreover, Manvar and collaborators demonstrated that luteolin can inhibit the RNA-dependent RNA polymerase (RdRp) activity of the non-structural protein 5B (NS5B) of hepatitis C virus with a IC_50_ equal to 52 µM [51]. More recently, the in vitro antiviral activity of luteolin against influenza A virus was outlined, specifically at the early stages of its lifecycle. Evidence that luteolin blocked the absorption and internalization of influenza virus was brought to light using time of addition assay. The highlighted mode of action of luteolin in the study was the one involving the host protein COPI [49].

To gain insight into the potential mechanism of action of luteolin against HCoV-229E, we added luteolin at different time points during infection and compared its activity with those of selected inhibitors, camostat, GC376 and remdesivir. Our data showed that luteolin had an antiviral activity at the same step as GC-376 (protease inhibitor) and remdesivir (RdRp inhibitor), when it was added at post-inoculation (1 h or 2 h). These findings indicated that luteolin is an inhibitor of the replication step of HCoV-229E. However, further experiments are needed to determine its precise mechanism of action.

One of the main challenges in the fight against HCoVs is to find broad-spectrum antivirals capable of inhibiting both alphacoronavirus and betacoronavirus. Luteolin turned out to be inactive against both SARS-CoV-2 and MERS-CoV in our in vitro experiments showing that luteolin did not have a broad-spectrum activity against all HCoVs. Its activity seemed to be specific to the alphacoronavirus HCoV-229E. It would be interesting, in the future, to test its capacity to inhibit other coronaviruses belonging to alphacoronavirus genus. Luteolin has been the subject of multiple in silico studies showcasing molecular docking simulations. Unlike our in vitro results, in silico studies highlighted a potential interest of luteolin against SARS-CoV-2 infections. It was described as an allosteric modulator of the spike protein of SARS-CoV-2 [56]. In another study, luteolin was also reported as a potent inhibitor that is capable of getting entry into the active sites of Mpro (3CLpro) and ACE2 [57]. In the same way, the inhibitory activity of luteolin against the RdRp was demonstrated in the experimental-computational study of Munafò and his collaborators [58]. However, these in silico strategies must remain complementary to the in vitro and in vivo approaches as they may have some limitations [59]. In silico analysis were not confirmed by our in vitro assays.

A study conducted by Yi et al., right after the global outburst of SARS-CoV in 2003, has identified luteolin, among other small molecules, as a potential drug, which can interfere with the entry of SARS-CoV into host cells. Thanks to the frontal affinity chromatography-mass spectrometry (FAC/MS) method, luteolin was selected because it exhibited a strong binding affinity to the fusion protein GST-S2, an S2 protein fragment that corresponds to the sequence between Asn733 to Gln1190 of the SARS-CoV S protein and plays a crucial role in the virus-cell fusion process. To investigate its antiviral activity, HIV-luc/SARS pseudotyped virus and wild-type SARS-CoV infection assays were performed on Vero E6. Respective EC_50_ values of 9.02 µM and 10.6 µM were determined for luteolin [50]. These results suggested that luteolin could be active at the entry pathway of SARS-CoV. Despite the similarities of SARS-CoV and SARS-CoV-2, we were not able to show an activity of luteolin on SARS-CoV-2 entry. It would be interesting to determine if luteolin is active on SARS-CoV, in a cell culture assay reproducing the complete viral cycle.

A better understanding of the mechanism of action of anti-SARS-CoV-2 candidates, including molecules from plant origin, is also highly important as natural products and/or their synthetic analogues can veritably accelerate the process of discovering and developing new molecular entities with greater efficacy and affinity, and with fewer side effects. The combination of already-existing antivirals with bioactive natural compounds such as luteolin or other drugs can also be a promising approach for developing anti-coronavirus treatments [22].

The bioavailability of luteolin still deserves further study, but it appears to be absorbed quite rapidly into plasma and persists in plasma after being metabolized to glucuronide and sulfate conjugates [60,61]. Luteolin was also previously identified as a highly bioactive phytochemical in *Artemisia afra* Jacq. ex Willd (Asteraceae), a plant widely used in traditional medicine especially for treating respiratory disorders. This flavone was demonstrated to contribute significantly in the broncho-dilatory effects of *Artemisia afra* notably under its nebulized form [62]. More recently, luteolin was also the focus of a clinical trial that aimed to treat olfactory dysfunction of SARS-CoV-2-affected patients. This latter showed that supplementation with palmitoylethanolamide (PEA) combined with luteolin can greatly improve the recovery of olfactory functions [63]. Luteolin is considered non-toxic with oral LD50 value greater than 2500 mg/kg in mouse and greater than 5000 mg/kg in rats [64]. No toxicity was observed in humans when administered at a dose of 100 mg/day [61]. Taken together, our data showed that luteolin might be a promising agent against mild HCoVs. Due to its good safety and its broad-spectrum antiviral activities, luteolin could provide a clinical solution for coronavirus treatment. Further experiments are needed to determine its precise mechanism of action.

## 4. Materials and Methods

### 4.1. Plant Material

Three *Juncus* species were collected in 2 regions in France, Brittany (2 different localities: 1 and 2) and Nouvelle-Aquitaine (locality 3) during July 2018 (Table 1 and Figure 12). Botanical identification was performed by Drs. Gabriel Lefèvre and Céline Rivière and a voucher specimen for each plant was deposited at the laboratory of Pharmacognosy, University of Lille, France.

### 4.2. Plant Extract Preparation

All collected plants were dried at 30 °C in a light protected environment. Their organs (rhizomes, stems and inflorescences) were powderized separately and stored in the dark. The crude extract of each organ was obtained by performing 4 sessions of maceration (12 h for each) in methanol, which were done at room temperature, away from light and under stirring. At the end of each maceration, the filtered methanolic extract was evaporated at 30 °C by a rotary evaporator (Heidolph™, Schwabach, Germany). The obtained dry methanolic extracts were then suspended in water in a round-bottom flasks, freezed overnight and finally lyophilized at −50 °C using a freeze dryer (Telstar Cryodos™, Barcelona, Spain) for better and longer preservation. The yield of each crude methanolic extract (%) was calculated based on the dry weight (20 g for rhizomes and stems and 10 g for inflorescences) (Table 2).

### 4.3. Solvent-Solvent Partition of Juncus acutus Stems (JA2 SCE)

Only the crude extract of *Juncus acutus* stems (5.15 g) was subjected to partitioning with liquid-liquid extraction using solvents with increasing polarity. The crude extract was suspended in water (300 mL) and the partitioning was, firstly, performed with methylene chloride (CH_2_Cl_2_) (Carlo Erba Reagents^®^, Val de Reuil, France) (3 × 300 mL) then with ethyl acetate (EtOAc) (Carlo Erba Reagents^®^, Val de Reuil, France) (3 × 300 mL). Anhydrous sodium sulfate (Na_2_SO_4_) was added to the organic phase in order to remove residual traces of water. After filtration, the CH_2_Cl_2_ phase and the EtOAc phase were evaporated using a rotary evaporator and the aqueous phase was frozen then lyophilized. The yield percentages of the three obtained sub-extracts were the following: 7.13% for the CH_2_Cl_2_ sub-extract, 4.54% for the EtOAc sub-extract and 72.58% for the aqueous sub-extract.

### 4.4. Fractionation of the EtOAc Sub-Extract of Juncus acutus Stems

The fractionation of the EtOAc sub-extract of *Juncus acutus* stems was performed using semi-preparative HPLC. The equipment was composed of Shimadzu^®^ LC-20AP binary high-pressure pumps, a SPD-M20A photodiode array detector and a CBM-20A controller. A VisionHT C_18_ HL (5 μm, 250 × 10 mm) column (Grace, batch number 60/036) was used in this experiment as the stationary phase. The mobile phase was composed of ultra-pure water (Millipore Integral 5 Milli-Q, Merck™, Trosly-Breuil, France) + 0.1% formic acid (Merck™, Darmstadt, Germany) (solvent A) and acetonitrile (Carlo Erba Reagents^®^, Val de Reuil, France) (solvent B). The elution program started with 5% of solvent B to reach 100% within 60 min. The flow rate was set at 3 mL·min^−1^. 8 injections of 12.5 mg of EtOAc sub-extract of *Juncus acutus* stems solubilized in 500 µL of MeOH were performed. The fractionation monitoring was carried out at three main wavelengths: 254 nm, 280 nm and 320 nm. 12 fractions (EtOAc F1 → EtOAc F12) were obtained at the end of the experiment. They were, later on, analyzed with UHPLC-UV-MS.

### 4.5. Fractionation of the Aqueous Sub-Extract of Juncus acutus Stems

The fractionation of the aqueous sub-extract of *Juncus acutus* stems was carried out by centrifugal partition chromatography (CPC). Several ternary systems were tested in different proportions: AcOEt/ACN/H_2_O, AcOEt/Isopropanol/H_2_O and AcOEt/Butanol/H_2_O. The ternary system composed of EtOAc/Isopropanol/H_2_O (7:3:10; *v*/*v*/*v*) was selected because it provided a partition coefficient (Kd) for luteolin-7-*O*-glucoside close to 1 (Kd = 1.022), whereas luteolin remained in the upper phase.

The CPC instrument (Armen instruments^®^, Saint-Avé, France) was equipped with a column compartment (250 mL rotor), a Shimazu^®^ pump (LC-20AP, Kyoto, Japan), a DAD detector (SPD-M20A) and an automated fraction collector (Gilson^®^ FC 204, Villiers-le-Bel, France). The elution profile was monitored by LabSolutions™ software version 1.25.

The CPC rotor was firstly filled with the stationary phase (lower phase) at a flow rate of 30 mL·min^−1^ (500 rpm) in ascending mode. Equilibration was attained while introducing the mobile phase (upper phase) at 1600 rpm and a flow rate of 8 mL·min^−1^. 1 g of the aqueous sub-extract of *Juncus acutus* stems, previously dissolved in 8 mL of the organic/aqueous phase mixture (1:1, *v*/*v*) and filtered through a Millipore (0.45 μm) syringe filter, was immediately injected after the displacement of stationary phase (100 mL). The elution was done at 8 mL·min^−1^ for a duration of 60 min and monitored at λ = 254 nm. After that, extrusion mode was performed to allow recover the molecules that were heavily retained in the stationary phase. At the end of the CPC run, the 105 obtained fractions were characterized by UHPLC-UV-MS and then pooled into 12 fractions according to their phytochemical profiles. The 12 fractions were then concentrated by a centrifugal concentrator (Genevac™, Fisher-Scientific, Illkirch, France)

### 4.6. UHPLC-UV-MS Analysis

The Acquity UPLC H-Class Waters^®^ System (Guyancourt, France) apparatus was equipped with two independent pumps, a controller, a diode array detector (DAD) and a QDa electrospray quadrupole mass spectrometer. The stationary phase was a C_18_ BEH (2.1 × 50 mm, 1.7 µm) reverse column. The mobile phase was composed of two solvents: (A) ultrapure water + 0.1% formic acid (Carlo Erba Reagents^®^, Val de Reuil, France) and (B) Acetonitrile (Carlo Erba Reagents^®^, Val de Reuil, France) + 0.1% formic acid. A method was specially developed for the analysis of *Juncus* extracts, sub-extracts, fractions and purified compounds. Flow rate and column temperature were set at 0.3 mL.min^−1^ and 30 ± 5 °C respectively. Wavelength range was fixed at 200–790 nm with a resolution of 1.2 nm. Ionization was carried out in both negative and positive mode with a mass range of 100 to 950 Da. Cone voltage and capillary voltage values were 15 V and 0.8 kV respectively. Injection volume was set at 2 µL. UHPLC-UV-MS analysis were executed following the elution program: 10% → 100% (B) (0–9 min), 100% (B) (9–11 min) and 10% (B) (11–14 min). All samples were prepared at 1 mg·mL^−1^ in analytical grade MeOH and filtered through a PTFE 0.4 µm membrane before injection. Only standards, luteolin (Sarsyntex), luteolin-7-*O*-glycoside (Sarget) and phenanthrene derivatives such as juncusol (purified in the laboratory) were prepared at 0.1 mg·mL^−1^. The acquired data were compared to online databases like PubChem (https://pubchem.ncbi.nlm.nih.gov/) (accessed on 21 November 2022).

### 4.7. Cells and Culture Conditions

Human hepatoma cell line Huh-7 and African monkey kidney cell lines Vero-81 cells were grown in DMEM with glutaMAX-I and 10% FBS in an incubator at 37 °C with 5% CO_2_. Human lung cell line Calu-3 (ATCC number HTB-55) was maintained in MEM supplemented with 10% FBS and glutaMAX-I.

### 4.8. Viruses

The following viral strains were used: recombinant HCoV-229E-Luc (kindly gifted by Pr. V. Thiel) [65]; SARS-CoV-2 (isolate SARS-CoV-2/human/FRA/Lille_Vero-81-TMPRSS2/2020, NCBI MW575140) and MERS-CoV (MERS-CoV-EMC12, kindly provided by Luis Enjuanes).

### 4.9. Cell Toxicity Assay

6 × 10^4^ Huh-7 cells were seeded in 96-well plates and incubated for 16 h at 37 °C 5% CO_2_ incubator. The cells were then treated with increasing concentrations of each extract, fraction or compound and incubated at 37 °C 5% CO_2_ for 23 h. An MTS [3-(4,5-dimethylthiazol-2-yl)-5-(3-carboxymethoxyphenyl)-2-(4-sulfophenyl)-2H-tetrazolium]-based viability assay (Cell Titer 96 Aqueous non-radioactive cell proliferation assay, Promega) was performed as recommended by the manufacturer. The absorbance of formazan at λ = 490 nm was detected using a plate reader (ELX 808 Bio-Tek Instruments Inc., Charlotte, VT, USA). Each measure was performed in triplicate and each experiment was repeated at least 3 times.

### 4.10. Antiviral Activity Assay on HCoV-229E

Hepatoma cell line Huh-7 and Huh-7 cells transduced with a lentivirus encoding for TMPRSS2 protease, a cellular protease that allows fusion of the virus at the host cell surface (Huh-7/TMPRSS2), were used for all HCoV-229E infection assays. All the above-mentioned samples from *Juncus acutus* stems were screened for their antiviral activity against HCoV-229E-Luc expressing the luciferase, a recombinant HCoV-229E with luciferase reporter gene. Huh-7 cells seeded in 96-well plates were inoculated with HCoV-229E-Luc in the presence of each extract, fraction or compound for 1 h. Inoculum was then removed and replaced with culture medium containing the compounds for another 6 h. Finally, cells were lysed in 20 µL of Renilla lysis buffer (Promega) and luciferase activity was quantified using Renilla luciferase assay kit (Promega). Luciferase activity was measured by the use of a Tristar LB941 luminometer (Berthold Technologies, Bad Wildbad, Germany).

### 4.11. SARS-CoV-2 Infection Assay

A Vero-81 reporter cell line for SARS-CoV-2 infection, F1G-Red, was used in this study. F1G-red cells were seeded in 384-well plates at a concentration of 4000 cells per well. The next day, cells were inoculated with SARS-CoV-2 (MOI 0.2) in the presence of 50 nM tariquidar and increasing doses of luteolin (0, 1.56, 3.13, 6.25, 12.5 or 25 µM) or GC376 (0, 0.125, 0.250, 0.5, 1 or 2 µM). 16 h post-inoculation, image acquisition was performed using an InCell-6500 automated confocal microscope (Cytiva) and percentages of infection and total cell numbers were assessed as described before [66].

### 4.12. MERS-CoV Infection Assay

Calu-3 cells, seeded on coverslips 24 h before inoculation, were inoculated with MERS-CoV at an MOI of 0.2 in the presence of increasing doses of luteolin (0, 1.56, 3.13, 6.25, 12.5 or 25 µM) or remdesivir (0 or 5 µM). 16 h post-inoculation, cells were fixed with 4% PFA. After permeabilization with 0.1% Triton-X100 in PBS for 5 min at RT, cells were incubated with 10% normal goat serum for 20 min at RT. Viral dsRNA was visualized by incubation with mouse monoclonal anti-dsRNA antibodies (clone J2, Scicons) in 10% normal goat serum, followed by incubation with Alexa Fluor^®^ 488-conjugated goat-anti-mouse IgG secondary antibodies. Nuclei were visualized with 1 µg/mL of 4′,6-diamidino-2-phenylindole (DAPI), and coverslips were mounted in Mowiol^®^ mounting medium. Images were obtained with an EVOS M5000 imaging system and percentage of infected cells and total cell numbers were quantified with the Image J software version 1.5i.

## 5. Conclusions

Luteolin, isolated from *Juncus acutus* stems, showed a very strong antiviral activity against the alphacoronavirus HCoV-229E. Our study underlined that this ubiquitous flavone acts as an inhibitor of the replication step. The identification of its exact target could be a future perspective. On the other hand, the activity of luteolin could not be demonstrated against the betacoronaviruses, SARS-CoV-2 and MERS-CoV. This flavone is therefore not active against all of the members of the *Coronaviridae* family, suggesting that it could target a HCoV-229E protein(s) or a cellular factor specifically involved in HCoV-229E RNA replication. Moreover, our results are in discordance with some in silico studies focused on luteolin and highlight the relevance of using traditional in vitro strategies. To conclude, luteolin is showing promising results and could serve as a lead for further drug development in the treatment of coronavirus based on different strategies such as nanotechnology, pharmacomodulation, combination drug therapies or even a prodrug strategy [67].

## Figures and Tables

**Figure 1 molecules-28-04263-f001:**
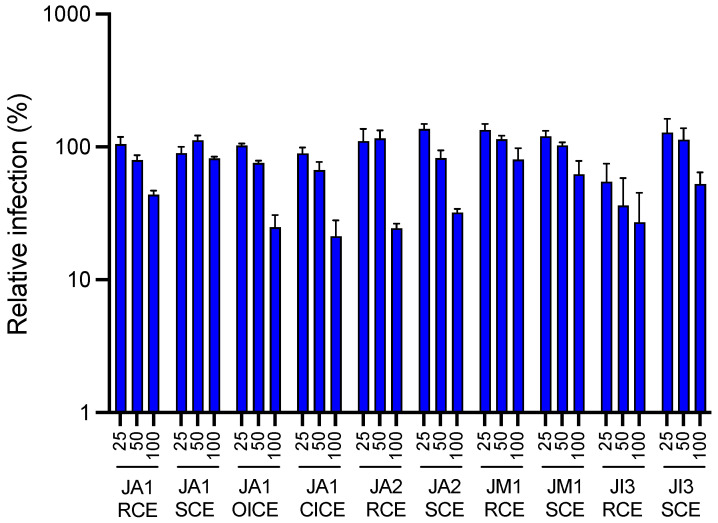
Antiviral activity of the different crude methanolic extracts of *Juncus* species against HCoV-229E-Luc in Huh-7/TMPRSS2 cells. Huh-7/TMPRSS2 cells were inoculated with HCoV-229E-Luc in the presence of the different extracts at 25, 50 and 100 µg·mL^−1^. The cells were lysed 7 h post-inoculation and luciferase activity was quantified. Data are expressed at mean ± SEM of 3 experiments performed in triplicate and relative to the control without extract (DMSO) for which a value of 100 was attributed. JA1 = *J. acutus* (locality 1); JA2 = *J. acutus* (locality 2); JM1 *J. maritimus* (locality 1); JI3 = *J. inflexus* (locality 3); RCE = rhizome crude extract; SCE = stem crude extract; OICE = open inflorescence crude extract; CICE = closed inflorescence crude extract.

**Figure 2 molecules-28-04263-f002:**
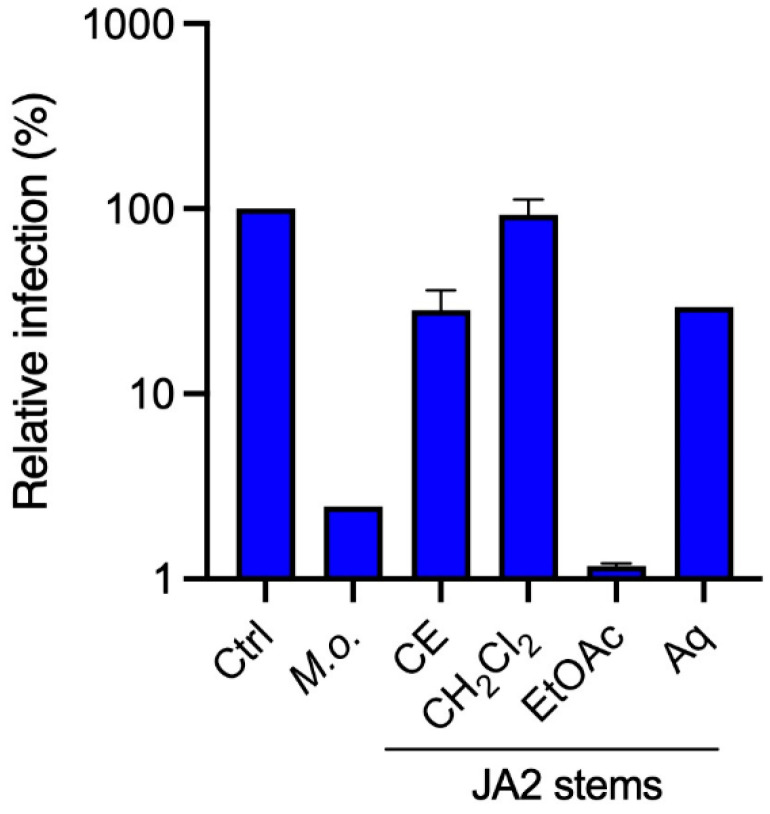
Antiviral activity of crude methanolic extract and sub-extracts obtained from JA2, *Juncus acutus* stems, against HCoV-229E-Luc in Huh-7/TMPRSS2 cells. Huh-7/TMPRSS2 cells were inoculated with HCoV-229E-Luc in the presence of the different extracts at 25 µg·mL^−1^. The cells were lysed 7 h post-inoculation and luciferase activity was quantified. Results are expressed at the mean ± SEM of one or two experiments performed in triplicate. Results are expressed relative to the control (DMSO).

**Figure 3 molecules-28-04263-f003:**
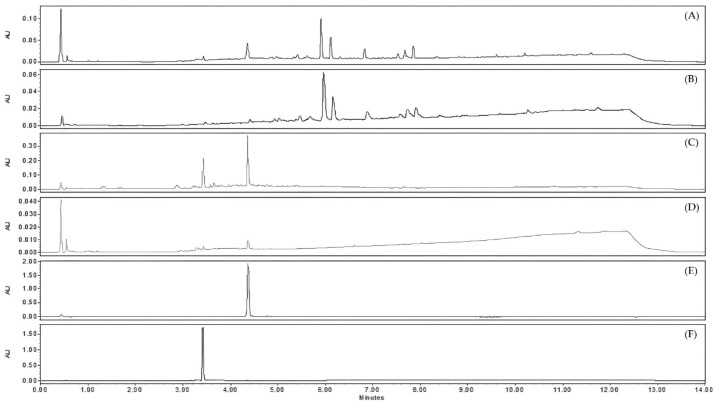
Chromatograms obtained by UHPLC-UV-MS at λ = 254 nm of different extracts of *J. acutus* stems (JA2) (**A**) crude methanolic extract, (**B**) CH_2_Cl_2_ sub-extract, (**C**) EtOAc sub-extract, (**D**) aqueous sub-extract; as well as reference standards (**E**) luteolin, (**F**) luteolin-7-*O*-glucoside.

**Figure 4 molecules-28-04263-f004:**
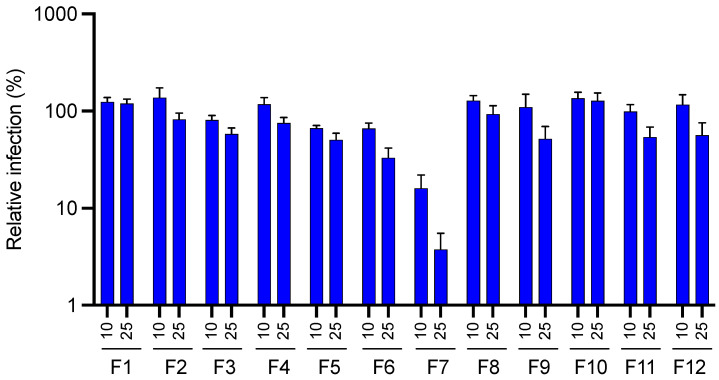
Anti-HCoV-229E activity of the different pre-purified fractions obtained from the EtOAc sub-extract of *J. acutus* stems (JA2) by semi-preparative HPLC. Huh-7/TMPRSS2 cells were inoculated with HCoV-229E-Luc in the presence of the different fractions (F1–F12) obtained from JA2 EtOAc sub-extract at 10 and 25 µg·mL^−1^. Luciferase activity was quantified 7 h post-inoculation. Data are presented relative to control DMSO for which a value of 100 was attributed and as mean ± SEM of 3 independent experiments performed in triplicate.

**Figure 5 molecules-28-04263-f005:**
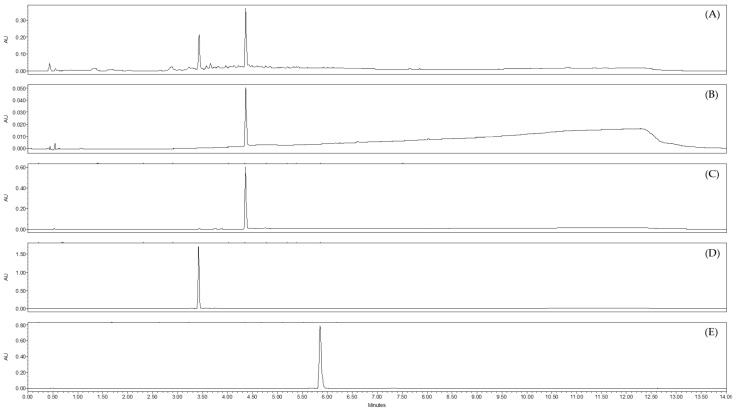
Chromatograms acquired by UHPLC-UV-MS at λ = 254 nm of (**A**) EtOAc sub-extract of *J. acutus* stems (JA2), (**B**) the active pre-purified fraction F7 obtained by semi-preparative HPLC from the same sub-extract, and reference standards (**C**) luteolin, (**D**) luteolin-7-*O*-glucoside and (**E**) juncusol (at λ = 280 nm).

**Figure 6 molecules-28-04263-f006:**
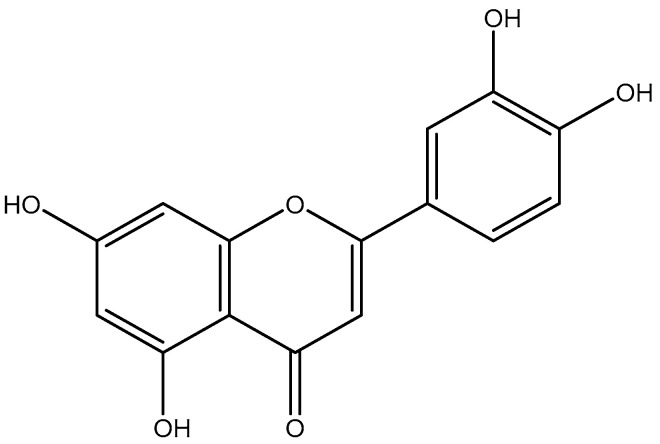
Luteolin.

**Figure 7 molecules-28-04263-f007:**
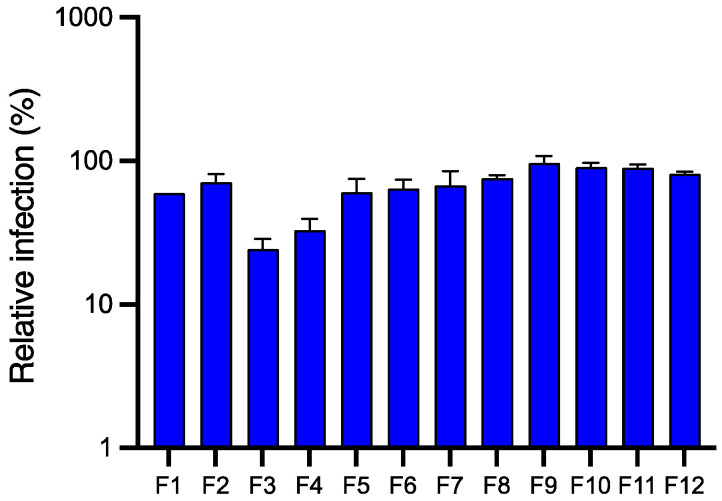
Anti-HCoV-229E activity of the different pre-purified fractions obtained from the aqueous sub-extract of *J. acutus* stems (JA2) by CPC. Huh-7/TMPRSS2 cells were inoculated with HCoV-229E-Luc in the presence of the different fractions (F1–F12) obtained from JA2 aqueous sub-extract at 50 µg·mL^−1^. Data are expressed relative to control (DMSO) and as mean ± SEM of 3 independent experiments performed in triplicates.

**Figure 8 molecules-28-04263-f008:**
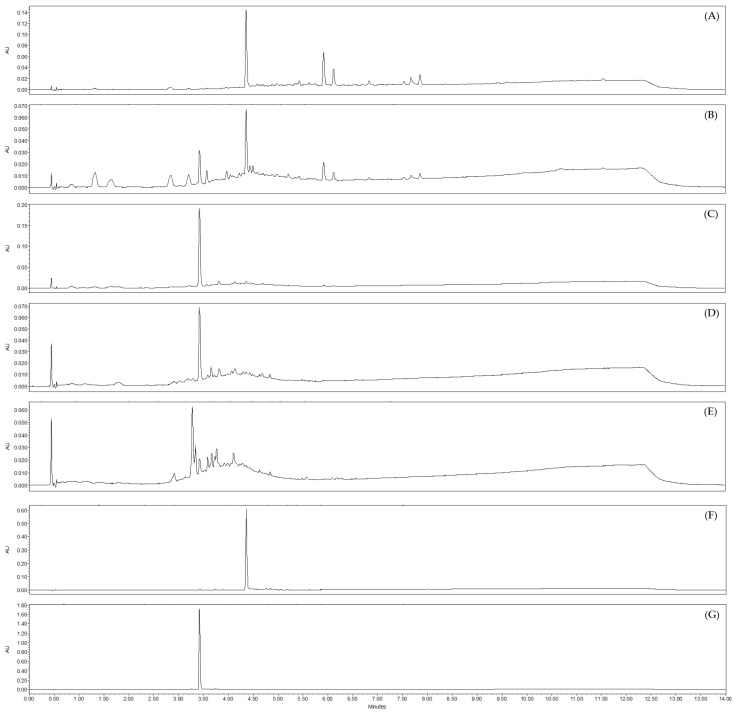
Chromatograms obtained by UHPLC-UV-MS at λ = 254 nm of the most active pre-purified fractions obtained by CPC from the aqueous sub-extract of *J. acutus* stems (JA2), (**A**) fraction F3, (**B**) fraction F4, as well as non-active fractions from the same sub-extract, (**C**) fraction F5, (**D**) fraction F6, (**E**) fraction F7. (**F**) luteolin and (**G**) luteolin-7-*O*-glucoside are reference standards.

**Figure 9 molecules-28-04263-f009:**
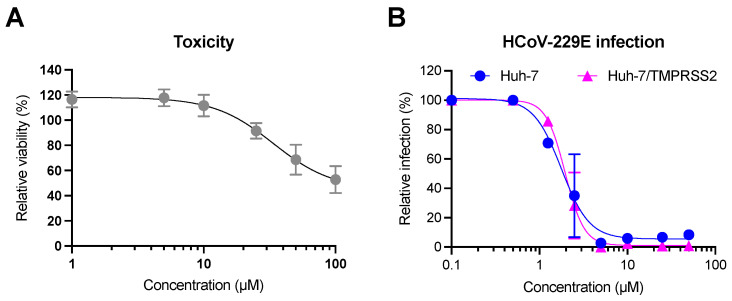
Toxicity and antiviral activity of luteolin. For toxicity assay (**A**), Huh-7 cells were incubated with different concentrations of luteolin for 24 h. The medium was replaced with MTS to determine the cell viability. For antiviral assays (**B**), Huh-7 cells expressing or not TMPRSS2 were inoculated with HCoV-229E-Luc in the presence of different concentration of luteolin for 1 h. The inoculum was removed and replaced with culture medium containing luteolin at the same concentration; 7 h post-inoculation, cells were lysed and luciferase activity was quantified. Results were expressed relative to the control for which a value of 100% (either viability or infection) was attributed. Results are the mean ± SEM of 3 experiments performed in triplicates.

**Figure 10 molecules-28-04263-f010:**
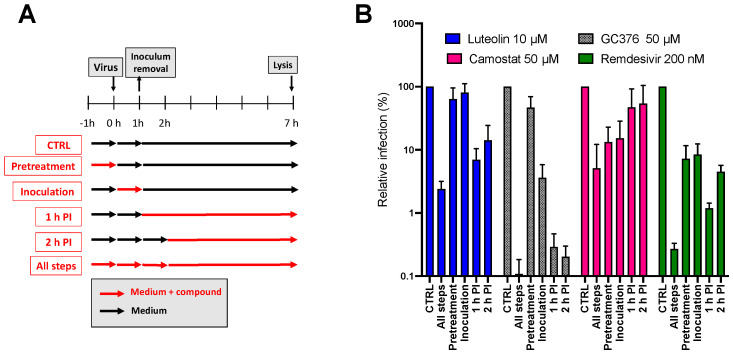
Luteolin is active at the replication step. (**A**) Schematic representation of the time of addition assay; (**B**) Huh-7/TMPRSS2 cells were inoculated with HCoV-229E-Luc. Luteolin (10 µM), GC376 (50 µM), camostat (50 µM) and remdesivir (200 nM) were added at different time points either before, during or post-inoculation (1 h or 2 h post-inoculation). Cells were lysed 7 h post-inoculation and luciferase activity was quantified. Results were expressed relative to the control without compound for which a value of 100 was attributed. Results are means ± SEM of 3 experiments performed in triplicate.

**Figure 11 molecules-28-04263-f011:**
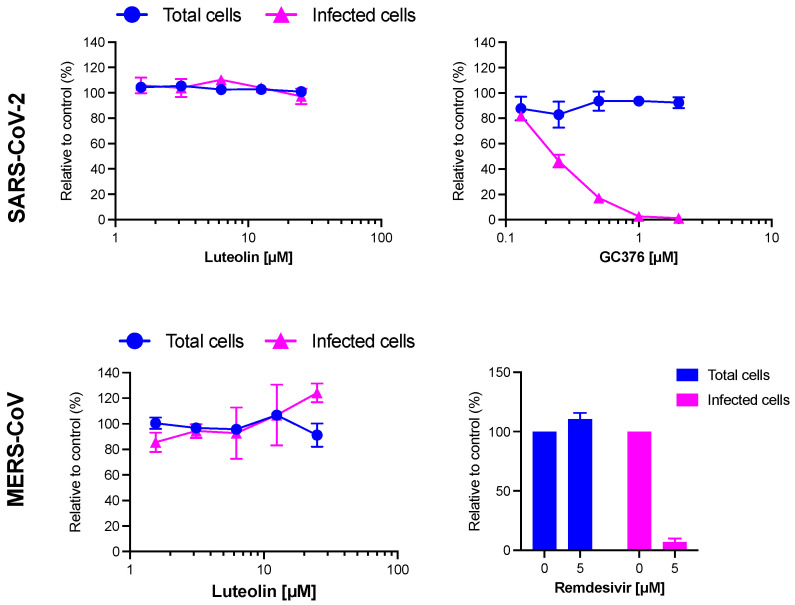
Luteolin is not active against SARS-CoV-2 and MERS-CoV. Vero81-derived F1G-red cells and Calu-3 cells were inoculated with SARS-CoV-2 and MERS-CoV (MOI 0.2), respectively, in the presence of increasing concentrations of luteolin or control compounds (GC376 and remdesivir for SARS-CoV-2 and MERS-CoV, respectively). For SARS-CoV-2 infection in F1Gred cells, infected cells and total cell numbers were recorded and quantified 16 h post-inoculation as previously described (Desmarets et al., 2022). For MERS-CoV infection in Calu-3 cells, cells were fixed with 4% PFA 16 h post-inoculation and immunofluorescence staining was performed against dsRNA. Nuclei were visualized with DAPI. Images were obtained with an EVOS M5000 imaging system and percentage of infected cells and total cell numbers were quantified with the Image J software version 1.5i. Data are presented as percentage of infection relative to the DMSO controls. Data are means ± SEM of 2 independent experiments performed in triplicates.

**Figure 12 molecules-28-04263-f012:**
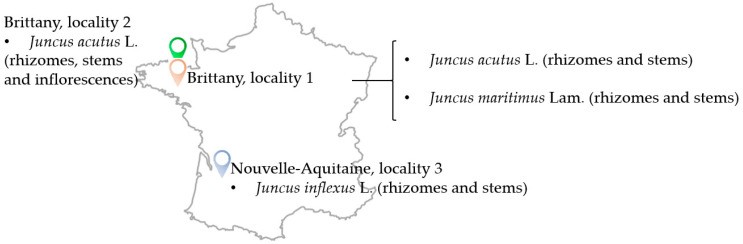
Illustrative map showing the locations from which *Juncus* species were collected during July 2018 (France).

**Table 1 molecules-28-04263-t001:** Collected *Juncus* species used in this study.

Species	Plant Part	Region	Geographical Coordinates
*Juncus acutus* L.	rhizomes, stems, inflorescences	Brittany, locality 1	48°48′44″ North, 3°05′27″ West
*Juncus acutus* L.	rhizomes, stems	Brittany, locality 2	48°49′48″ North, 3°04′40″ West
*Juncus inflexus* L.	rhizomes, stems	Nouvelle-Aquitaine, locality 3	44°59′44″ North, 0°26′41″ West
*Juncus maritimus* Lam	rhizomes, stems	Brittany locality 1	48°48′44″ North, 3°05′27″ West

**Table 2 molecules-28-04263-t002:** Yields of the different crude methanolic extracts obtained after solid/liquid extraction.

Species	Plant Part	Abbreviation	Yield (%)
*J. acutus* (locality 1)	rhizomes	JA1 RCE	9.6
	stems	JA1 SCE	12.3
	open inflorescences	JA1 OICE	2.7
	closed inflorescences	JA1 CICE	8.5
*J. acutus* (locality 2)	stems	JA2 SCE	11.85
	rhizomes	JA2 RCE	6.95
*J. maritimus* (locality 1)	rhizomes	JM1 RCE	11.25
	stems	JM1 SCE	11.6
*J. inflexus* (locality 3)	rhizomes	JI3 RCE	5.2
	stems	JI3 SCE	11.5

## Data Availability

Not applicable.

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
