# Peer review of "Luteolin Isolated from Juncus acutus L., a Potential Remedy for Human Coronavirus 229E"

_molecules, 2023, doi:10.3390/molecules28114263_

Round 1

Reviewer 1 Report

The article " Luteolin isolated from Juncus acutus L., a potential remedy for human coronavirus 229E" attempts to highlighting the potential value of Luteolin in combating HCoV-229E. The paper is well structured. I recommend this article to be published in the journal. Here are some suggestions:

1. Check the abbreviations throughout the manuscript. Make a word abbreviated in the article that is repeated at least two times in the text, not all words to be abbreviated (For example, HCoV-HKU1).

2. There is a lack of recent literature citations. For example, in lines 73-77, “To deal with the SARS-CoV-2 pandemic, the scientific community worldwide put into action many strategies to counteract SARS-CoV-2 infection, such as blocking SARS-CoV-2 entry with monoclonal antibodies (DOI: 10.3390/v14020255), impairing SARS-CoV-2 replication with polymerase or protease inhibitors (DOI: 10.3389/fphar.2022.926507; DOI: 10.1099/jmm.0.001203), inhibiting excessive inflammatory response by repurposing already existing antivirals and broad-spectrum drugs (DOI: 10.1002/jmv.27517; DOI: 10.1158/2159-8290.cd-21-0144).”

3. In “Conclusions” section, the quality of the Conclusions must be improved. For example, “Natural products have demonstrated potential value and with the assistance of nanotechnology, lead optimization, combination drug therapies, and the prodrug strategy, this “natural remedy” could serve as a starting point for further drug development in treating coronavirus. (DOI: 10.3390/biomedicines9060689)”

4. “Luteolin has shown excellent safety and broad‐spectrum antiviral activities that could contribute immediate clinical solution for coronavirus treatment.” For the benefits of the readers please list more detailed information.

Editing of English language required.

Author Response

Please see our reply in the attached file.

Reviewer 2 Report

Some clarification: Why just luteolin was followed in the extracts? What standards were used? Where the lypohilisated forms of the plant were used?

Why  you choose this solvents? "Even though their crude methanolic extracts demonstrated 155 the most important antiviral activity, it was difficult to continue working on them" Why?

Author Response

(The authors gave the same response as above.)
